# Resveratrol-Loaded *Attalea funifera* Oil Organogel Nanoparticles: A Potential Nanocarrier against A375 Human Melanoma Cells

**DOI:** 10.3390/ijms241512112

**Published:** 2023-07-28

**Authors:** Douglas Dourado, Fabiana Pacheco Reis Batista, Biane Oliveira Philadelpho, Myla Lôbo de Souza, Mariana Barros de Cerqueira e Silva, Rone Aparecido de Grandis, Priscila Anjos Miranda, Nelson Barros Colauto, Daniel T. Pereira, Fabio Rocha Formiga, Eduardo Maffud Cilli, Fernando Rogério Pavan, Carolina Oliveira de Souza, Ederlan de Souza Ferreira

**Affiliations:** 1School of Pharmacy, Federal University of Rio Grande do Norte (UFRN), General Gustavo Cordeiro de Faria Street, Natal 59012-570, RN, Brazil; danieltpereira19@gmail.com; 2Department of Immunology, Aggeu Magalhães Institute (IAM), Oswaldo Cruz Foundation (FIOCRUZ), Recife 50670-420, PE, Brazil; fabio.formiga@fiocruz.br; 3School of Pharmacy, Federal University of Bahia, Barão de Jeremoabo Street, Salvador 40170-115, BA, Brazil; fabianaprb@gmail.com (F.P.R.B.); biane_philadelpho@hotmail.com (B.O.P.); priscila_anj@yahoo.com.br (P.A.M.); nelsonbcolauto@gmail.com (N.B.C.); carolods@ufba.br (C.O.d.S.); 4College of Pharmacy, Federal University of Pernambuco, Professor Artur de Sá Street, Recife 50740-521, PE, Brazil; mylobo.souza@gmail.com; 5School of Pharmacy, Federal University of Amazonas, General Rodrigo Octávio Jordão Ramos Avenue, Manaus 69067-005, AM, Brazil; marianabarros@ufam.edu.br; 6School of Pharmacy, São Paulo State University (UNESP), Araraquara-Jaú Road, Araraquara 148000-903, SP, Brazil; degrandis.rone@gmail.com (R.A.d.G.); fernando.pavan@unesp.br (F.R.P.); 7Institute of Chemistry, São Paulo State University (UNESP), Prof. Francisco Swgni Street, Araraquara 14800-0600, SP, Brazil; eduardo.cilli@unesp.br

**Keywords:** piassava seed oil, medium-chain fatty acids, gelled lipid nanoparticles, cell viability, skin cancer

## Abstract

This study aimed to evaluate *Attalea funifera* seed oil with or without resveratrol entrapped in organogel nanoparticles in vitro against A375 human melanoma tumor cells. Organogel nanoparticles with seed oil (SON) or with resveratrol entrapped in the seed oil (RSON) formed functional organogel nanoparticles that showed a particle size <100 nm, polydispersity index <0.3, negative zeta potential, and maintenance of electrical conductivity. The resveratrol entrapment efficiency in RSON was 99 ± 1%. The seed oil and SON showed no cytotoxicity against human non-tumor cells or tumor cells. Resveratrol at 50 μg/mL was cytotoxic for non-tumor cells, and was cytotoxic for tumor cells at 25 μg/mL. Resveratrol entrapped in RSON showed a decrease in cytotoxicity against non-tumor cells and cytotoxic against tumor cells at 50 μg/mL. Thus, SON is a potential new platform for the delivery of resveratrol with selective cytotoxic activity in the treatment of melanoma.

## 1. Introduction

Resveratrol is a polyphenol found in peanuts, red grape bark, and red wine, and is a promising multi-target anticancer agent with the potential of being used in various tumor chemoprevention and chemotherapies [1]. Particularly in cutaneous melanoma, which is considered an aggressive skin cancer that often causes treatment recurrence and has a high mortality rate with increasing worldwide incidence [2], resveratrol inhibits the growth, proliferation, and apoptosis of melanoma cells via different signaling pathways [3]. However, it has a low aqueous solubility and bioavailability, which reduce its anticancer activity [4]. To overcome these limitations, topical drug delivery systems have been used to improve the penetration and permeation of low water-soluble drugs such as resveratrol into the skin [5,6].

Nanoparticles have been used to increase drug [7,8,9,10,11] and nutraceutical [6,12,13] deposition in the target region to enhance physicochemical stability and promote the sustained and controlled delivery of molecules such as resveratrol [4,5,14,15,16,17]. Moreover, lipid-based nanoparticles have attractive and versatile biological characteristics [8] such as biocompatibility, biodegradability, and the ability to entrap hydrophilic and hydrophobic substances [9] when compared to other nanostructures containing polymers and/or the use of organic solvents in their production [8,18]. Solid lipid nanoparticles and nanostructured lipid nanoparticles have been used for melanoma treatment [6,19]. However, they have critical limitations such as low encapsulation capacity and drug expulsion after polymorphic transition during storage [20]. Gelled lipid nanoparticles have been developed to overcome the limitations of these nanostructures in terms of greater physical stability, encapsulation, entrapping capacity, and cell internalization, mainly due to the combination of particle and gel properties at the nanoscale [21]. When a low-molecular-mass gelling agent is used to structure the organic phase of the gel, it is referred to as an organogel, configuring the nanocarrier as organogel nanoparticles [22].

Organogel nanoparticles have been used with 12-hydroxystearic acid (12-HSA) as the preferred gelling agent [23,24] because it is commercial gelling, biocompatible, and can form a gel with various organic solvents and oils [25]. This gelling agent is capable of structuring a lipid phase at much lower concentrations [23] than the lipids used for the formation of conventional nanoparticles [18].

In addition, medium-chain triglyceride oils are of interest for their incorporation into organogels as they confer high stability, skin permeation, and enhanced sensory characteristics compared to unsaturated fatty acid oils [26]. Furthermore, medium-chain triglyceride oils contain a high concentration of saturated fatty acids (C8:0 to C12:0) [27], which are associated with anti-inflammatory and moisturizing properties [28] as well as antioxidant [29] and anticancer skin activity [30].

*Attalea funifera* is a palm tree endemic to Brazil and is cultivated to provide a source of fiber to manufacture brooms and handicrafts [31]. It contains oil seeds, and the oil from these seeds has not yet been reported or characterized. However, the seed oils from *Attalea speciosa*, *Attalea dubia*, and *Attalea phalerata* have been reported to contain medium-chain triglyceride oils [32], and therefore, a similar fatty acid profile is expected to be found in the seed oil of *A. funifera*. Hence, this study aimed to characterize and evaluate *A. funifera* seed oil with and without resveratrol entrapped in organogel nanoparticles against in vitro human non-tumor and tumor cells.

## 2. Results and Discussion

### 2.1. Attalea funifera Seed Oil Characterization

The seed oil of *A. funifera* presented low acidity and peroxide index (Appendix A) compared to the recommended limits of 4.0 mg KOH/g for cold-pressed oils and 15 mEq/kg for unrefined oils [33]. The refractive index (20 °C) (Appendix A) is an important parameter for quality control as it increases with an increase in molecular mass and fatty acid saturation [34]. To the best of our knowledge, no study has compared the physicochemical characteristics of *A. funifera* seed oil; therefore, this is the first study to report the characteristics of this oil.

The seed oil contained more than 90% saturated fatty acids and lauric and myristic acids constituted the major part (70%) (Table 1). It was also found to contain medium-chain fatty acids such as capric, caprylic, and lauric acids. The fatty acid composition of *A. funifera* seed oil is consistent with that reported for palm species and is a rich source of medium-chain saturated fatty acids. Oils rich in medium-chain fatty acids are used as lubricants, emollients, antimicrobials, and anti-inflammatory agents in foods, pharmaceuticals, and cosmetics [9,28]. Thus, the chemical composition of the oil reported in this study hints at interesting future applications.

### 2.2. Organogel Characterization

SO (seed oil organogel) and RSO (resveratrol-entrapped in the seed-oil organogel) were obtained. The inverted test tube assay is shown in the Appendix A (Appendix A). It was verified that there was no extravasation of oil from the gel, confirming that the seed oil in SO and resveratrol-seed oil in RSO were entrapped in the gel, which can therefore be termed as functional organogels. The addition of resveratrol in the organogel changed the transition initial temperatures from 40.1 ± 0.4 °C to 39.2 ± 0.6 °C (solidification temperature) and 50.5 ± 1.6 °C to 48.0 ± 0.9 °C (melting temperature) without statistical differences (*p* > 0.05) between the solidification or melting temperature values. The temperature reduction indicated the formation of a three-dimensional network. Moreover, the melting temperature of approximately 50 °C is still higher than the normal skin temperature of 37 °C [35], which would allow the organogel to be used on the skin without melting.

#### 2.2.1. Particle Size and Polydispersity Index

The particle size remained constant for 90 days and ranged from 84.93 ± 0.35 to 86.09 ± 0.24 nm for SON (seed oil organogel nanoparticles; 1.3% variation) and from 75.72 ± 0.65 to 76.19 ± 0.93 nm for RSON (resveratrol-seed oil organogel nanoparticles; 0.6% variation) (Table 2). The size heterogeneity and stability verified by the polydispersity index were also constant and ranged from 0.266 ± 0.009 to 0.273 ± 0.004 for SON (2.6% variation) and from 0.264 ± 0.004 to 0.282 ± 0.004 for RSON (6.4% variation) (Table 2). For drug delivery, particle sizes lower than 200 nm [36] and a polydispersity index lower than 0.3 [37] are desired, and hence, SON and RSON can enhance drug permeability and retention in the tumor cells [36], resulting in better anti-tumor treatment [38].

The zeta potential for 90 days ranged from −26.27 ± 1.00 to −31.57 ± 1.76 mV for SON (16.8% variation) and −21.90 ± 0.35 to −32.27 ± 1.00 mV for RSON (32% variation) (Table 3). In general, the negative charge on nanoparticles indicates low cytotoxic potential [39]. At 90 days, a zeta potential of −31.57 ± 1.76 mV for SON and −32.27 ± 1.00 mV for RSON indicated a high electrostatic repulsion and greater stability of the organogel nanoparticles [39]. The stability of the nanoparticles in suspension is complex. Mechanisms such as steric and/or electrostatic stabilization are involved in the colloidal stability of these suspensions [40]. In the organogel nanoparticles of our study, we initially observed a steric stabilization by the surfactant chains. The increase in charge (in modulus) provides greater stability through electrostatic stabilization combined with the steric stabilization already mentioned. As the colloidal medium is a dynamic medium, there is an intense exchange of ions between the particles, their components, and the dispersing medium. The balance between interactions and electrostatic repulsions is described by DLVO’s (Derjaguin, Landau, Verway, Overbeek) theory [41].

#### 2.2.2. pH Analysis and Electrical Conductivity

The pH remained somewhat constant for 90 days, ranging from 6.03 ± 0.07 to 6.23 ± 0.04 for SON (3% variation) and from 5.99 ± 0.04 to 6.56 ± 0.05 for RSON (9% variation) (Table 4). The increase in pH over time can be indicative of the degradation process of the surfactant. The surfactant polysorbate 80 (Tween^®^ 80), in starting the process of auto-oxidation of the ethylene group, releases organic acids, consequently, there is a reflection in the increase in the pH of the formulation [42,43]. The pH became more acidic than that at the beginning, but was still compatible with the pH of the human skin [44,45]. Different formulations intended for this route are also found in this acidic pH range [46,47,48]. Additionally, no visual changes were detected in the organogel nanoparticles.

The electrical conductivity over the 90 days ranged from 101 ± 0.47 to 100.53 ± 0.42 µs/cm for SON and from 130.20 ± 0.50 to 130.00 ± 0.66 µs/cm for RSON, without potential chemical degradation (Table 5). This indicates that resveratrol in RSON increased the electrical conductivity values compared to SON (*p* < 0.001). A small fraction of resveratrol (approximately 1%) was not entrapped in the core of the nanoparticles and was dispersed in the aqueous phase. The OH^−^ groups present in this molecule are susceptible to protonation in the media (suspension pH 6), as shown in the Appendix A (Appendix A). This suggests that H^+^ ions are released in the suspension, leading to greater electrical conductivity.

#### 2.2.3. Resveratrol Entrapment Efficiency in Organogel Nanoparticles

The resveratrol entrapment efficiency in the organogel nanoparticles with seed oil was 99 ± 1%, which corresponds to 20 mg/g oil. A high entrapment of resveratrol in the organogel nanoparticles was seen when compared to other resveratrol-entrapped in lipid nanoparticles such as solid lipid nanoparticles (80.5% equivalent to 2 mg/g) and nanostructured lipid carriers (78.9% equivalent to 1.97 mg/g oil) [17]. Low entrapment efficiency values (55–67%) in conventional lipid nanoparticles have also been reported in the entrapment of other anticancer molecules [49]. The observed entrapping capacity of the organogel nanoparticles was closely related to the self-assembled three-dimensional fibrillar networks of the organogel formed due to the physical interactions between the organogelator agent 12-HSA, oil phase, and drug. This behavior has also been observed in other structured organogel nanoparticles with the same gelling agent, which showed an entrapping capacity greater than 90% [10].

#### 2.2.4. Fourier-Transform Infrared Spectroscopy Analysis

Fourier-transform infrared spectroscopy (FTIR) analyses of SON, RSON, and isolated components such as *Attalea funifera* seed oil, resveratrol, and 12-HSA were performed to investigate the entrapment of resveratrol by the nanoparticles (Figure 1).

The SO spectra revealed that absorptions in the 3000–2800 cm^−1^ range were due to C–H stretching vibrations that generally occur in fats and oils [50]. In addition, C=O stretching vibrations of esters (1743.65 cm^−1^) and C–H stretching vibrations of the out-of-plane cis-CH deformation (approximately 3000 cm^−1^) were detected [51]. Furthermore, the peaks observed at 2852 and 2922 cm^−1^ refer to functional groups –C–H(CH_3_) and –C–H(CH_2_) [51]. This spectrum is compatible with other medium-chain triglyceride oils [52,53]. According to the resveratrol FTIR results, it was possible to observe characteristic peaks around 1587 cm^−1^, representing the benzene vibration. The peaks around 830 cm^−1^ represent the bending vibration of C=C–H. In addition, an absorption band at 3200~3500 cm^−1^ (corresponding to the OH functional groups) is characteristic of this drug [16]. Moreover, the FTIR of 12-HSA showed a characteristic C=O bond stretching band at 1694 cm^−1^, assigned to the carboxylic acid, and there was axial stretching of C–H at 2845~3000 cm^−1^ [24]. The FTIR spectra of SON and RSON revealed a peak at 1745 corresponding to the C=O stretching vibrations of the esters (1743.65 cm^−1^) as well as the peaks in the region of 2800–3000 (C–H stretching vibrations) also observed in SON. Finally, the absorption band at 3200~3500 cm^−1^ corresponded to the OH functional groups. The similarity between the spectra of SON and RSON, without the presence of the characteristic peaks of resveratrol, suggests the entrapment of this drug in the nanoparticle, corroborating the high entrapment efficiency observed in our study (Section 2.2.3).

#### 2.2.5. Organogel Nanoparticle Morphology Analysis by Transmission Electron Microscopy and Size Distribution by Dynamic Light Scattering

Transmission electron microscopy (TEM) is a powerful tool to provide further information about the morphology of nanostructured systems such as lipid nanoparticles [54,55]. The nanoparticles showed spherical morphology with a rough diameter of 100 nm, close to the values found in the dynamic light scattering (DLS) analysis (Figure 2). Other organogel nanoparticles have also been reported with spherical morphology [25,56]. As observed in the histogram of the formulations, both particles showed a wide distribution due to the polydispersity index of these nanoparticles of about 0.3 (Table 2), which may reflect the small size difference observed in the DLS analysis when compared to the TEM (Figure 2).

An important highlight is that the techniques used have different fundamentals. TEM is based on the interaction between electron beams of uniform current density and a negatively stained thin sample. When the electron beam hits the deposited sample, some of the electrons are scattered as part of the electrons are transmitted, so an image is formed from these transmitted electrons [57,58]. On the other hand, dynamic light scattering (DLS) provides the hydrodynamic diameter, using mathematical models to generate the data displayed as particle size and to infer size distribution [22]. Based on this understanding, the two techniques have been used in combination to obtain a more robust measure [59].

### 2.3. Cell Viability Assay for Organogel Nanoparticles and Their Compounds

The seed oil of *A. funifera* and seed oil entrapped in SON promoted a cell viability close to 100% without differences (*p* > 0.05) at all tested concentrations when compared to the negative control. The positive control showed the lowest cell viability values of 57.9 ± 4.6% for HaCaT and 42.8 ± 4.2% for A375 cells (Figure 3). This shows that the seed oil, alone or entrapped in SON, is biocompatible with non-tumor (HaCaT) and tumor (A375) cells, as cell viability values ≥70% are assumed to be non-cytotoxic [60]. Moreover, at the highest concentration (500 μg/mL), SON showed a cell viability of 98.7 ± 6.8% and 94.0 ± 5.1% for the HaCaT cells, indicating biocompatibility with the topical formulation.

Resveratrol at concentrations from 50 to 500 μg/mL was cytotoxic to non-tumor (HaCaT) cells and cytotoxic to the tumor (A375) cells from 25 to 500 μg/mL (Figure 3). Resveratrol was more cytotoxic than the positive control at concentrations from 100 to 500 μg/mL for the HaCaT cells and 25 to 500 μg/mL for the A375 cells (Figure 3). At 500 μg/mL, resveratrol yielded a cell viability of 37.7 ± 6.5% for the HaCaT and 8.3 ± 2.1% A375 cells. The cytotoxicity profile of resveratrol has previously been reported [61,62]. Resveratrol was cytotoxic at concentrations ranging from 25 to 500 μg/mL, and the proposed mechanism of action involves the induction of apoptosis mediated by the expression of pro-apoptotic proteins [63].

The resveratrol at concentrations from 6.25 to 500 μg/mL in RSON showed biocompatibility with non-tumor (HaCaT) cells with cell viability values of 80.3 ± 8.0% at 500 μg/mL (Figure 3), showing low toxicity. This reduction in chemo-drug cytotoxicity in non-tumor cells has been observed in nanosystems such as nanoparticles, given the targeting provided by these nanoparticles. Furthermore, the physiological lipid composition of lipid-based nanoparticles plays an important role in the biocompatibility of these nanosystems [11].

On the other hand, RSON was cytotoxic against the A375 tumor cells at concentrations ranging from 50 to 500 μg/mL, although it caused lower inhibition (*p* < 0.001) compared to the same concentrations of resveratrol without the presence of organogel nanoparticles. The least cellular inhibition by the nanoparticles occurred due to the entrapment of resveratrol inside the organogel nanoparticles, thus requiring a longer time to deliver this drug to the cell membrane. Although RSON was cytotoxic to the tumor (A375) cells at concentrations higher than or equal to 50 μg/mL, it was biocompatible with the non-tumor (HaCaT) cells at all concentrations assessed. These findings suggest that the cytotoxicity of RSON may involve a different underlying molecular mechanism in tumor cells.

The selectivity index (SI) of each treatment was calculated. The SI reflects a compound’s selectivity by comparing the IC_50_ in non-cancer cells to the IC_50_ in cancer cells. The SI calculation of resveratrol and RSON was 7.58 and 2.96, respectively (Table 6). It has been reported that nanoparticles commonly show a greater uptake in tumor cells than in non-tumor cells, which leads to selective cytotoxicity [15,64].

## 3. Materials and Methods

### 3.1. Chemicals

The chemicals used in the assay were as follows: Tween^®^ 80 (T1029, Labsynth, Diadema, Brazil); Liovac^®^ 1112 [saturated fatty acid derived from hydrogenated castor oil, containing 88% pure 12-hydroxystearic acid (12-HSA), 12% oleic acid (C18), and palmitic acid (C16)] (Miracema-Nuodex, Campinas, Brazil); resveratrol (98% purity; Bothânica, Araraquara, Brazil); alamarBlue^TM^, a cell viability reagent with resazurin (DAL1025, Thermo Fisher Scientific, Waltham, MA, USA); trypan blue solution (0.4%; 15250061; Gibco^TM^, Waltham, MA, USA); DMEM (Dulbecco’s modified Eagle’s medium/high modified) (51441C; Merck, Darmstadt, Germany); fetal bovine serum (Gibco^TM^); trypsin-EDTA (ethylenediaminetetraacetic acid) solution (T4049, Sigma-Aldrich, St. Louis, MO, USA); DMSO (dimethyl sulfoxide) (≥99.5% purity, D4540, Sigma-Aldrich); methyl methanesulfonate (99% purity, 129925, Sigma-Aldrich).

### 3.2. Plant Material and Seed Oil Extraction

*Attalea funifera* Mart. (Arecaceae), endemic to the Atlantic rainforest, is commonly found in the northeast and southeast of Brazil, and is popularly known as *piaçava* in Brazil; it has synonyms such as *Attalea acaulis* Burret, *Lithocarpos cocciformis* O.Targ.Tozz. ex Steud. and *Sarinia funifera* (Mart.) O.F.Cook [65].

The fruits of *A. funifera* were collected at 13°73′39.9″ S, 39°14′65.4″ W, in the Ituberá region (Bahia, Brazil). Undamaged and parasite-free fruits were chopped in half using a machete to obtain whole seeds. Seed mass was measured using a semi-analytical scale (Quimis^®^, Diadema, Brazil), and the seeds were stored at 4 °C until oil extraction.

For oil extraction, seeds were dried at 55 °C in an oven with air circulation for 12 h, ground in a blender, and crushed at 30 °C in a manually operated hydraulic system (Marconi^®^ model ME098, Piracicaba, Brazil). The oil extracted from the seeds was centrifuged (3420× *g* for 10 min at 10 °C) and stored in a closed amber glass container at 4 °C until analysis.

### 3.3. Physicochemical Characterization and Fatty Acid Composition of the Seed Oil

The acidity, peroxide, refractive indices, and relative density of *A. funifera* seed oil were determined using established methods [66]. The fatty acid profile was determined by gas chromatography (Clarus^®^ 680, PerkinElmer, MO, USA) with a flame ionization detector (GC-FID), and lipids were extracted using the Bligh–Dyer method [67]. The fatty acids were then transmethylated using hexane boron trifluoride. The fatty acid methyl esters (FAME) were analyzed using a GC-FID and fused silica capillary column DB-FFAP (30 m × 0.32 mm × 0.25 mm). The temperature parameter for the injector was 250 °C and for the detector, it was 280 °C. FAME was subjected to a thermal program of 150 °C for 16 min at an increasing rate of 2 °C/min until 180 °C was reached. This temperature was maintained for 25 min, then increased at a rate of 5 °C/min to 210 °C, which was then maintained for another 25 min. Helium was used as a carrier gas at a flow rate of 1.0 mL/min; the flow rate of hydrogen gas was 30 mL/min, and that of air was 300 mL/min. The injections (1 μL) were performed in duplicate for each extraction. FAME was identified by comparing the retention times with the standards (C4-C24, 18,919-AMP, Sigma-Aldrich, St. Louis, MO, USA). The peak areas were determined using TotalChrom^®^ Workstation software (version 6.3.2) to normalize the percentage areas of the total fatty acids [68]. The results were averaged from triplicate extractions.

### 3.4. Organogel Preparation and Characterization

Seed oil (0.97 g) and 12-HSA (0.03 g), a low molecular weight gelator agent, were mixed in a flask using a magnetic stirrer hot plate until melting at 90 °C and then cooled to 25 °C. The obtained seed oil organogel was denoted as SO. Resveratrol (20 mg/g), seed oil (0.97 g), and 12-HSA (0.03 g) were mixed in a flask on a magnetic stirrer hot plate until melting at 90 °C and cooled to the same temperature (25 °C). The resveratrol-entrapped in the seed-oil organogel was denoted as RSO. SO and RSO organogels were maintained at room temperature (25 °C).

Organogel formation was verified using the inverted test tube method [69]. The tubes with SO and RSO were inverted for 24 h, and the samples without any flow of the compound were assumed to be functional organogels. Visual analysis of the organogel transition temperature was carried out in heating and cooling cycles [70]. Each organogel was melted at 90 °C to assess the gel–sol transition temperature (melting temperature). After complete melting, the sample was left to cool to room temperature (25 °C) to assess the sol–gel transition temperature (solidification temperature). The assay was performed in triplicate on a heating plate (Z671797, IKA^®^ C-MAG HS, Sigma-Aldrich, St. Louis, MO, USA) coupled with a digital thermometer (Z645125, IKA^®^ ETS-D5, Sigma-Aldrich, St. Louis, MO, USA).

### 3.5. Organogel Nanoparticles Preparation and Characterization

#### 3.5.1. Organogel Nanoparticles Preparation

Organogel nanoparticles were prepared according to the method described [71] with minor modifications. Reverse osmosis water (20 mL) and Tween^®^ 80 (5% tween per mL water) were heated at 75 °C, poured into a tube with 1 g of SO or RSO, kept in an oven at 75 °C (above solidification temperature) for 20 min, and sonicated at 75 °C and 42 kHz for 20 min in an ultrasonic cleaner (Cristofoli^®^, Campo Mourão, Brazil). The dispersions were kept at room temperature (25 °C); the organogel nanoparticles of the SO dispersions were denoted as SON, and those of the RSO dispersions were denoted as RSON.

#### 3.5.2. Organogel Nanoparticles Size Distribution and Zeta Potential

The particle size and polydispersity index of SON and RSON were evaluated by dynamic light scattering (DLS) using a Zetasizer^®^ Nano (ZS90, Malvern-Panalytical, Malvern, UK) at 25 °C and a fixed angle of 90° for 90 days [22,59]. Additionally, the zeta potential was measured during the same period using the electrophoretic mobility technique with the same equipment. The organogel nanoparticles were diluted in purified water (1:20; *v*/*v*).

#### 3.5.3. Organogel Nanoparticles pH Analysis and Electrical Conductivity

The electrical conductivity and pH values of the organogel nanoparticles were obtained (n = 3) over 90 days using a conductivity meter (MC-226, Mettler Toledo^®^, Columbus, OH, USA) and a pH meter (TEC-2, Tecnal^®^, Piracicaba, Brazil) at 25 °C [22,48].

#### 3.5.4. Resveratrol-Entrapment Efficiency of Organogel Nanoparticles

The organogel nanoparticles were centrifuged (HEA10050, Gusto^®^ high-speed, Heathrow Scientific, IL, USA) at 9800× *g* for 15 min. The supernatant was diluted in ethanol, and the resveratrol content was quantified using a UV–VIS spectrophotometer (Libra S32; Biochrom^®^, Cambridge, UK) at 306 nm with a calibration curve (y = 1.3844x + 0.0249; r^2^ = 0.9999). The resveratrol-entrapment efficiency in the organogel nanoparticles was calculated as the total resveratrol mass in the organogel nanoparticles subtracted from the resveratrol-free mass in the supernatant, divided by the total resveratrol mass in the organogel nanoparticles, and multiplied by 100. The result was expressed as a percentage (%).

#### 3.5.5. Morphological Study by Transmission Electron Microscopy (TEM)

The morphology of the organogel nanoparticles was observed by transmission electron microscopy (TEM) (FEI Tecnai G2 Spirit Biotwin, 120 kV, FEI Company, Hillsboro, OR, USA). The suspensions of nanoparticles were diluted in purified water (1:20, *v*/*v*) and negatively stained with an uranyl acetate aqueous solution (2%) for 1 min. A drop of the nanoparticle suspension was fixed over Cooper square mesh grids (2 min). The samples were dried at room temperature [22,58,59].

### 3.6. Cell Viability Assay of Organogel Nanoparticles and Their Compounds

The cell viability of the human non-tumor cell line HaCaT (human cultured keratinocyte cell line—BCRJ0341) and human tumor cell line A375 (human malignant melanoma epithelial-skin cell line—BCRJ0278) was determined by the resazurin redox assay [72] with minor modification [73]. Cells were incubated under standard cell culture conditions (37 °C, 5% CO_2_, 95% humidity, and 24 h) containing DMEM culture medium supplemented with fetal bovine serum (10 g/100 mL) in 96-well microplates (1.5 × 10^4^ cells/well).

For the treatment, samples were dissolved in DMSO (5 mL/L) to obtain solutions from 6.25 to 500 µg/mL. The negative control (NC) consisted of cells that received the culture medium and DMSO (5 mL/L) and the positive control (PC) consisted of cells that received the culture medium and the cytotoxic agent methyl methanesulfonate (MMS, 300 µM). These were incubated in the same standard cell culture conditions for 24 h. Then, the culture medium was removed, and 50 µL of 0.01% resazurin hydrochloride (Sigma-Aldrich^®^, Merck, St. Louis, MO, USA) was added. The plates were again incubated for 2 h, then the fluorescence was read in a Synergy H1 microplate reader (BioTek^®^, Hongkong, China) in excitation and emission filters at 530 and 590 nm, respectively. All experiments were performed in triplicate. The results were expressed by the percentage of cell viability calculated based on the fluorescence emitted by the negative control, considered as 100% viability. The selectivity index (SI) was calculated using the following formula: SI = IC_50_ [non-tumor cells]/IC_50_ [tumor cells]. IC_50_ is the half inhibitory concentration of a sample or drug.

### 3.7. Statistical Analyses

All assays were carried out in triplicate, and the results expressed as the arithmetic average ± standard deviation. Significant differences were assessed using the analysis of variance (ANOVA) and Student’s *t*-test (*p* ≤ 0.05, 0.01, and/or 0.001) or Tukey’s multiple range test (*p* ≤ 0.05). The analyses were carried out with GraphPad^®^ Prism 5 software (GraphPad Holdings, San Diego, CA, USA).

## 4. Conclusions

Organogel nanoparticles with *A. funifera* seed oil were established as a successful new platform for resveratrol delivery with low aqueous solubility. SON and RSON formed functional organogel nanoparticles with melting temperatures above that of the human skin. In addition, high-efficiency resveratrol entrapment was observed in the organogel nanoparticles. These organogel nanoparticles (SON and RSON) showed stability in terms of the evaluated parameters (particle size, polydispersity index, zeta potential, pH, and electrical conductivity) over 90 days. Resveratrol alone at 50 μg/mL was cytotoxic for non-tumor cells, and was cytotoxic for tumor cells at 25 μg/mL. In contrast, resveratrol entrapped in RSON had no cytotoxicity against non-tumor cells and it was cytotoxic against A375 cells at 50 μg/mL with a SI of 2.96. Finally, RSON showed selective cytotoxic activity with potential applications in melanoma treatment. However, further in vitro and in vivo studies are needed to elucidate the underlying molecular mechanisms.

## Figures and Tables

**Figure 1 ijms-24-12112-f001:**
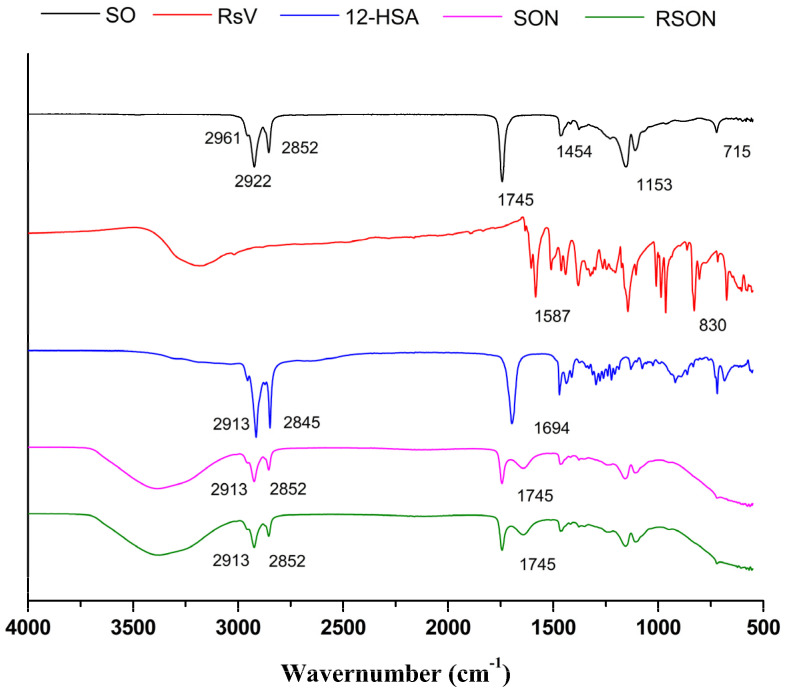
FTIR analyses of the *Attalea funifera* seed oil (SO), resveratrol (RsV), 12-hydroxystearic acid (12-HSA), *Attalea funifera* seed oil organogel nanoparticles (SON), and resveratrol entrapped in *Attalea funifera* seed oil organogel nanoparticles (RSON).

**Figure 2 ijms-24-12112-f002:**
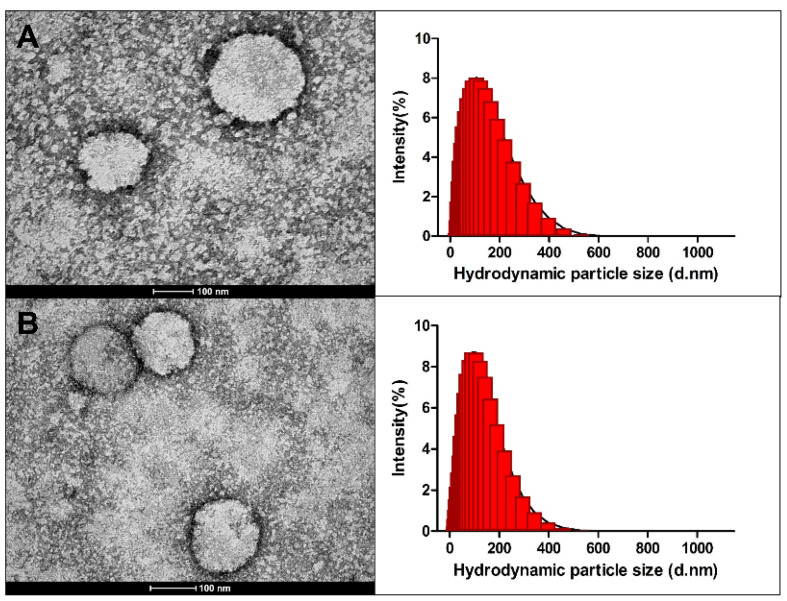
Morphology and intensity of the (**A**) *Attalea funifera* seed oil organogel nanoparticles (SON) and (**B**) resveratrol entrapped in *Attalea funifera* seed oil organogel nanoparticles (RSON) by transmission electron microscopy (TEM) and the size distribution histogram by dynamic light scattering (DLS), respectively.

**Figure 3 ijms-24-12112-f003:**
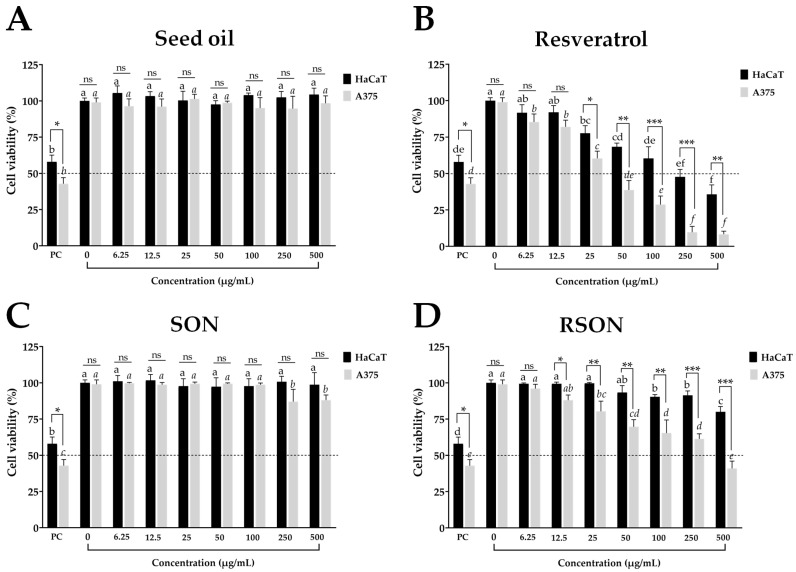
Cell viability (%) of the human non-tumor (HaCaT) and tumor (A375) cell lines at different concentrations of (**A**) *Attalea funifera* seed oil, (**B**) resveratrol, (**C**) *Attalea funifera* seed oil organogel nanoparticles (SON), and (**D**) resveratrol entrapped in *Attalea funifera* seed oil organogel nanoparticles (RSON). Results are presented as the arithmetic mean ± standard deviation (*n* = 3). Cell viabilities of the negative control were 100 ± <0.1%, solvent control 96.3 ± 2.5%, and the positive control was 31.8 ± 9.6%. A cell viability ≥70% was assumed to be non-cytotoxic. Different letters for the same cell line indicate statistical differences according to Tukey’s multiple range test (*p* < 0.05). Comparison between the HaCaT and A375 cells at a given concentration, with asterisks (* *p* < 0.05, ** *p* < 0.01, and *** *p* < 0.001) indicate significant differences or with “ns” indicates no significant differences according to the Student’s *t*-test. PC = positive control with methyl methanesulfonate (300 µM); HaCaT = human cultured keratinocyte cell line; A375 = human malignant melanoma epithelial-skin cell line. The highlighted dotted line represents 50% cell viability.

**Table 1 ijms-24-12112-t001:** Fatty acid composition of *Attalea funifera* seed oil obtained by gas chromatography *.

Compound	Value (%)
Caprylic acid (C8:0)	1.96 ± 0.07
Capric acid (C10:0)	8.32 ± 0.30
Lauric acid (C12:0)	55.29 ± 1.93
Myristic acid (C14:0)	15.13 ± 0.42
Palmitic acid (C16:0)	6.17 ± 0.47
Stearic acid (C18:0)	3.26 ± 0.04
Oleic acid (C18:1 ∆^9^ cis) ω-9	8.21 ± 0.22
Linoleic acid (C18:2 ∆^9.12^ cis) ω-6	1.64 ± 0.04
Total saturated fatty acid	90.13
Total monounsaturated fatty acid	8.21
Total polyunsaturated fatty acid	1.64

* Arithmetic average ± standard deviation (*n* = 3).

**Table 2 ijms-24-12112-t002:** Particle size distribution and polydispersity index of the seed oil organogel nanoparticles (SON) and resveratrol-seed oil organogel nanoparticles (RSON) at 25 °C for 90 days *.

Time (Day)	SON		RSON	
	Particle Size (nm)	Polydispersity Index	Particle Size (nm)	Polydispersity Index
01	86.09 ± 0.24 ^aA^	0.273 ± 0.004 ^aA^	76.19 ± 0.93 ^aB^	0.278 ± 0.002 ^aA^
15	84.93 ± 0.35 ^aA^	0.271 ± 0.004 ^aB^	76.42 ± 0.34 ^aB^	0.282 ± 0.004 ^aA^
30	85.63 ± 0.33 ^aA^	0.266 ± 0.009 ^aA^	75.72 ± 0.65 ^aB^	0.264 ± 0.004 ^bA^
60	85.59 ± 0.30 ^aA^	0.268 ± 0.006 ^aA^	75.91 ± 0.09 ^aB^	0.262 ± 0.002 ^bA^
90	85.81 ± 0.59 ^aA^	0.272 ± 0.008 ^aA^	75.92 ± 0.04 ^aB^	0.266 ± 0.002 ^bA^

* Values expressed as the arithmetic average ± standard deviation (*n* = 3). Different lowercase letters in the same column indicate statistical differences against the values obtained on the first day, and different uppercase letters indicate statistical differences between the formulations at the same time (day), both according to the Student’s *t*-test. The *p*-value considered was <0.05.

**Table 3 ijms-24-12112-t003:** The zeta potential of *Attalea funifera* seed oil organogel nanoparticles (SON) and resveratrol-seed oil organogel nanoparticles (RSON) at 25 °C for 90 days.

Time (Day)	Zeta Potential (mV) *	
	SON	RSON
01	−26.27 ± 1.00 ^aA^	−21.90 ± 0.35 ^aB^
15	−23.50 ± 0.44 ^bA^	−23.63 ± 0.58 ^aA^
30	−21.13 ± 0.06 ^bA^	−23.60 ± 0.80 ^aA^
60	−23.94 ± 0.06 ^bA^	−23.53 ± 1.19 ^aA^
90	−31.57 ± 1.76 ^bA^	−32.27 ± 1.00 ^bA^

* Values expressed as the arithmetic average ± standard deviation (*n* = 3). Different lowercase letters in the same column indicate statistical differences against the values obtained on the first day, and different uppercase letters indicate statistical differences between the formulations at the same time (day), both according to the Student’s *t*-test. The *p*-value considered was <0.05.

**Table 4 ijms-24-12112-t004:** pH value of the *Attalea funifera* seed oil organogel nanoparticles (SON) and resveratrol-seed oil organogel nanoparticles (RSON) at 25 °C for 90 days.

Time (Day)	pH *	
	SON	RSON
01	6.03 ± 0.07 ^aA^	5.99 ± 0.04 ^aA^
15	6.19 ± 0.06 ^bB^	6.30 ± 0.06 ^bA^
30	6.20 ± 0.01 ^bB^	6.33 ± 0.01 ^bA^
60	6.23 ± 0.04 ^bB^	6.44 ± 0.06 ^bA^
90	6.22 ± 0.02 ^bB^	6.56 ± 0.05 ^bA^

* Values expressed as the arithmetic average ± standard deviation (*n* = 3). Different lowercase letters in the same column indicate statistical differences against the values obtained on the first day, and different uppercase letters indicate statistical differences between the formulations at the same time (day), both according to the Student’s *t*-test. The *p*-value considered was <0.05.

**Table 5 ijms-24-12112-t005:** Electrical conductivity of *Attalea funifera* seed oil organogel nanoparticles (SON) and resveratrol-seed oil organogel nanoparticles (RSON) at 25 °C for 90 days.

Time (Day)	Electrical Conductivity (µs/cm) *	
	SON	RSON
01	101.47 ± 0.21 ^aB^	130.20 ± 0.50 ^aA^
15	97.33 ± 0.23 ^bB^	121.53 ± 0.31 ^bA^
30	96.07 ± 0.23 ^bB^	122.60 ± 0.44 ^bA^
60	92.10 ± 0.44 ^bB^	122.23 ± 0.42 ^bA^
90	100.53 ± 0.42 ^aB^	130.00 ± 0.66 ^aA^

* Values expressed as the arithmetic average ± standard deviation (*n* = 3). Different lowercase letters in the same column indicate statistical differences against the values obtained on the first day, and different uppercase letters indicate statistical differences between the formulations at the same time (day), both according to the Student’s *t*-test. The *p*-value considered was <0.05.

**Table 6 ijms-24-12112-t006:** Concentration capable of inhibiting 50% of cell growth (IC_50_) and selectivity index (SI) of *Attalea funifera* seed oil, resveratrol, *Attalea funifera* seed oil organogel nanoparticles (SON), and resveratrol entrapped in *Attalea funifera* seed oil organogel nanoparticles (RSON) on human non-tumor (HaCaT) and tumor (A375) cells.

Treatment	IC_50_ (µg/mL) *		SI
	HaCaT	A375	
*Attalea funifera* seed oil	>500	>500	-
Resveratrol	238.86 ± 2.65	31.51 ± 1.41	7.58
SON	>500	>500	-
RSON	1250.00 ± 1.90	423.00 ± 1.21	2.96

* Values expressed as the arithmetic average ± standard deviation (n = 3). HaCaT = human cultured keratinocyte cell line; A375 = human malignant melanoma epithelial-skin cell line.

## Data Availability

Not applicable.

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
