# Peer review of "Resveratrol-Loaded Attalea funifera Oil Organogel Nanoparticles: A Potential Nanocarrier against A375 Human Melanoma Cells"

_ijms, 2023, doi:10.3390/ijms241512112_

Round 1

Reviewer 1 Report (Previous Reviewer 2)

Having read the revised manuscript I would like to thank the authors for the corrections made.  Apart from this I also have the following comments:

1.  Tables 3, 4, 5 and 6 how many replicates does the data shown within represent, is it 3 or another number.  Please add this information in the legend for these tables.  

2. Is the citation for Ref 23 correct?

3. Please correct reference numbers 39, 40, 46, 53, they appear to be incomplete.

The English used in the manuscript is appropriate and does not need significant editing apart from checking for typographical errors.

Author Response

Reviewer 1 – Comments and Suggestions for Authors

Having read the revised manuscript, I would like to thank the authors for the corrections made.  Apart from this I also have the following comments:

Answer: We would like to thank you for your criticisms and considerations, which contributed positively to the scientific quality of the manuscript. We have carefully considered all points rose and have modified the manuscript according to their comments. All changes within the original text were done in blue font. Please, see below the response point-to-point:

Point 1 – Tables 3, 4, 5 and 6 how many replicates does the data shown within represent, is it 3 or another number.  Please add this information in the legend for these tables.

Answer: The number of replicates has been added as requested. 

Point 2 – Is the citation for Ref 23 correct?

Answer: Several databases were consulted. The citation was adjusted to:

Citation 23 is now 33:

Codex Alimentarius Commission. Codex standard for named vegetable oils (Codex-Stan 210-1999), Amended 2005, 2011, 2013,2015). FAO/WHO, Rome, 2013; pp 1–13. Available in: https://inspection.canada.ca/DAM/DAM-food-aliments/WORKAREA/DAM-food-aliments/text-texte/codex_food_stand_named_veg_oils_1532975057193_eng.pdf

Point 3 – Please correct reference numbers 39, 40, 46, 53, they appear to be incomplete.

Answer: The references were adjusted to:

Reference 39 is now 54:

Elham A.; Rostamabadi, H.; Jafari, S.M. Chapter One - Characterization of Nanoencapsulated Food Ingredients. In In Nanoencapsulation in the Food Industry, Characterization of Nanoencapsulated Food Ingredients; Jafari, S.M., Ed.; Academic Press, 2020; pp. 1–50, ISBN 9780128156674, doi: https://doi.org/10.1016/B978-0-12-815667-4.00001-8.

Reference 40 is now 55:

Chen, C.; Chen, C.; Li, Y.; Gu, R.; Yan, X. Characterization of lipid-based nanomedicines at the single-particle level. Fundam. Res. 2023, 3, 488–504, doi:https://doi.org/10.1016/j.fmre.2022.09.011.

Reference 46 is now 60:

ISO-10993-5. Part 5 - Tests for in vitro cytotoxicity (ISO 10993-5:2009). In Biological evaluation of medical devices; International Organization for Standardization, 3rd Ed.; International Standard, 2009; pp. 1–34. Available in: https://nhiso.com/wp-content/uploads/2018/05/ISO-10993-5-2009.pdf

Reference 53 is now 65:

Govaerts, R. World Checklist of Selected Plant Families. In Catalogue of Life Checklist (Aug 2017); Bánki, O.; Roskov, Y.; Vandepitte, L.; DeWalt, R.E.; Remsen, D.; Schalk, P.; Orrell, T.; Keping, M.; Miller, J.; Aalbu, R.; Adlard, R.; Adriaenssens, E.; Aedo, C.; Aescht, E.; Akkari, N.; Alonso-Zarazaga, M. A.; Alvarez, B.; Alvarez, F.; Anderson, G. et al., 2017, doi: https://doi.org/10.48580/d4sd-38c.

We hope the revised version of the manuscript meets your expectations!

Reviewer 2 Report (New Reviewer)

The current manuscript aims to develop resveratrol-loaded Attalea funifera oil organogel nanoparticles for anticancer applications. In particular, the authors focus on the evaluation of nanoparticle as a potential nanocarrier against A375 human melanoma cells. Although the topic is significant in its scientific field, there are some issues that require the authors’ attention to improve the quality of this particular manuscript before further consideration for publication in a high-quality journal “IJMS”.

Specific comments:

1.         The authors should carefully clarify the differences in the academic contribution points between the current manuscript and the earlier reports involving the encapsulation of seed oil in the nanoparticles for skin cancer treatment (please refer to the following papers: #1 DOI: 10.1016/j.ejps.2015.08.016 & #2 DOI: 10.1016/j.jphotobiol.2015.03.007 & #3 DOI: 10.1016/j.bbrc.2018.11.106).

2.         As stated by the authors, the zeta potential for 90 days ranged from -21.13 to -31.57 mV for SON and -21.9 to -32.27 mV for RSON (Table 3). However, the numerical values reported in Table 3 are positive zeta potential rather than negative ones. Please carefully check the data presentation again. Furthermore, the standard deviations described in the text are inconsistent with tabular data. Please confirm.

3.         The authors should enrich the discussion by giving explanations about the decrease in zeta potential with increasing time (Table 3).

4.         As stated by the authors, the pH became more acidic than that at the beginning, but was still compatible with the pH of the human skin. Please specify the reason of increasing acidity at longer time points.

5.         Furthermore, the authors should provide literature report to support the claim that the acidity of carrier materials was compatible with the pH of the human skin.

6.         As stated by the authors, resveratrol in RSON increased the electrical conductivity values compared to SON (Table 5). The authors should enrich the discussion by giving relevant explanations about this finding.

7.         In the research background, the authors stated that nanoparticles have been used to increase drug [7] and nutraceutical [8,9] deposition in the target region to enhance physicochemical stability and promote sustained and controlled delivery. In fact, a recent report has also demonstrated the use of targeting nanoparticles for resveratrol (i.e., drug) delivery (please refer to DOI: 10.1021/acsnano.2c05824). The authors are highly recommended to consider the inclusion of this relevant case study in the reference list to enrich the article content and balance scientific viewpoint.

Author Response

Reviewer 2 – Comments and Suggestions for Authors

The current manuscript aims to develop resveratrol-loaded Attalea funifera oil organogel nanoparticles for anticancer applications. In particular, the authors focus on the evaluation of nanoparticle as a potential nanocarrier against A375 human melanoma cells. Although the topic is significant in its scientific field, there are some issues that require the authors’ attention to improve the quality of this particular manuscript before further consideration for publication in a high-quality journal “IJMS”. Specific comments:

Answer: We would like to thank you for your criticisms and considerations, which contributed positively to the scientific quality of the manuscript. We have carefully considered all points rose and have modified the manuscript according to their comments. All changes within the original text were done in blue font. Please, see below the response point-to-point:

Point 1 – The authors should carefully clarify the differences in the academic contribution points between the current manuscript and the earlier reports involving the encapsulation of seed oil in the nanoparticles for skin cancer treatment (please refer to the following papers: #1 DOI: 10.1016/j.ejps.2015.08.016 & #2 DOI: 10.1016/j.jphotobiol.2015.03.007 & #3 DOI: 10.1016/j.bbrc.2018.11.106).

Answer: Dear reviewer, thanks for these important references.

The references DOI: 10.1016/j.bbrc.2018.11.106 and DOI: 10.1016/j.ejps.2015.08.016 were inserted in lines 52-53:

Moreover, lipid-based nanoparticles have attractive and versatile biological characteristics [8] such as biocompatibility, biodegradability, and the ability to entrap hydrophilic and hydrophobic substances [9] when compared to other nanostructures containing polymers and/or use of organic solvents in their production [8,18].”.

Note: These exemplify nanostructured systems containing lipids that use organic solvents in their preparation or contain the addition of polymers for the formation of nanocapsules. On the other hand, the organogel nanoparticles developed by us are solvent-free, without the use of polymers, proving to be more physiologically biocompatible, with lower cost, and friendly to the environment.

The reference DOI: 10.1016/j.ejps.2015.08.016 was added to the text in lines 64-66:

This gelling agent is capable of structuring a lipid phase at much lower concentrations [23] than the lipids used for the formation of conventional nanoparticles [18]”.

Note: It was used to compare the composition of these conventional lipid nanostructures with organogel nanoparticles. Organogel nanoparticles require a low amount of low molecular weight gelling agent for the formation of the organogel particle, when compared to nanostructures such as SLNs, NLCs.

The reference DOI: 10.1016/j.jphotobiol.2015.03.007 was added to the text in lines 188-190:

 Low entrapment efficiency values (55-67%) in conventional lipid nanoparticles have also been reported in the entrapment of other anticancer molecules [49].

Note: This study reveals the production of 5-FU –NLCs, where was by authors observed a low entrapping efficiency of this molecule in NLCS. This limitation has already been pointed out for conventional lipid nanoparticles. From this limitation came the organogel nanoparticles which have a self-assembled three-dimensional configuration based on the interaction of the gelling agent and the lipid phase, which provide high efficiency for entrapping molecules.

Point 2 – As stated by the authors, the zeta potential for 90 days ranged from -21.13 to -31.57 mV for SON and -21.9 to -32.27 mV for RSON (Table 3). However, the numerical values reported in Table 3 are positive zeta potential rather than negative ones. Please carefully check the data presentation again. Furthermore, the standard deviations described in the text are inconsistent with tabular data. Please confirm.

Answer: Dear reviewer, thanks for this point. The information has been corrected in lines 130 to 131 and 168 to 169.

Point 3 – The authors should enrich the discussion by giving explanations about the decrease in zeta potential with increasing time (Table 3).

Answer: In order to enrich the discussion, the following information was added to the results and discussion section (lines 135 to 142):

“The stability of nanoparticles in suspension is complex. Mechanisms such as steric and/or electrostatic stabilization are involved in the colloidal stability of these suspensions [40]. In the organogel nanoparticles of our study, we initially observed a steric stabilization by the surfactant chains. The increase in charge (in modulus) provides greater stability through electrostatic stabilization combined with steric stabilization already mentioned. As the colloidal medium is a dynamic medium, there is an intense exchange of ions between the particles, their components and the dispersing medium. The balance between interactions and electrostatic repulsions is described by DLVO's (Derjaguin, Landau, Verway, Overbeek) theory [41]”.

Point 4 – As stated by the authors, the pH became more acidic than that at the beginning, but was still compatible with the pH of the human skin. Please specify the reason of increasing acidity at longer time points.

Answer: The information about the reason of increase acidity at longer time points was added to the results and discussion section (lines 154 to 160):

“The increase in pH over time can be indicative of degradation process of the surfactant. The surfactant polysorbate 80 (Tween ® 80), in starting the process of auto-oxidation of the ethylene group, releases organic acids, consequently, there is a reflection in the increase of the pH of the formulation [42,43]. The pH became more acidic than that at the beginning but was still compatible with the pH of the human skin [44,45]. Different formulations intended for this route are also found in this acidic pH range [46–48]. Additionally, no visual changes were detected in the organogels nanoparticles”.

Point 5 – Furthermore, the authors should provide literature report to support the claim that the acidity of carrier materials was compatible with the pH of the human skin.

Answer: This suggestion was clarified in the text as mentioned in the previous point.

Note: Dear reviewer, variable skin pH values are being reported in literature, all in the acidic range but with a broad range from pH 4.0 to 7.0 depending on the body part. [5-6]. The nanoparticles developed by us are in this range, being compatible with the desired application. Different formulations intended for this route are also found in this pH range (7-10).

Point 6 – As stated by the authors, resveratrol in RSON increased the electrical conductivity values compared to SON (Table 5). The authors should enrich the discussion by giving relevant explanations about this finding.

Answer: In order to enrich the discussion about the electrical conductivity values, the following information was added to the results and discussion section (lines 171 to 176):

A small fraction of resveratrol (approximately 1%) was not entrapped in the core of the nanoparticles and is dispersed in the aqueous phase. The OH- groups present in this molecule are susceptible to protonation in the media (suspension pH 6), as shown Supplementary Material (Figure S2). Thus, it is suggested that H+ ions are released in the suspension, leading to greater electrical conductivity”.

In addition, a scheme of protonation of the resveratrol was added as Supplementary Material (Figure S2).

Fig. S2 Scheme of protonation of the resveratrol observed at acid (pH 6) medium (MarvinSketch®, v. 18.11).

Point 7 – In the research background, the authors stated that nanoparticles have been used to increase drug [7] and nutraceutical [8,9] deposition in the target region to enhance physicochemical stability and promote sustained and controlled delivery. In fact, a recent report has also demonstrated the use of targeting nanoparticles for resveratrol (i.e., drug) delivery (please refer to DOI: 10.1021/acsnano.2c05824). The authors are highly recommended to consider the inclusion of this relevant case study in the reference list to enrich the article content and balance scientific viewpoint.

Answer: Dear reviewer, thanks for this reference. The reference DOI: 10.1021/acsnano.2c05824 was inserted in line 49.

We hope the revised version of the manuscript meets your expectations!

Reviewer 3 Report (Previous Reviewer 1)

Acceptable revisions have been made. The manuscript should be accepted.

Author Response

Reviewer 3 – Comments and Suggestions for Authors

Point 1 – Acceptable revisions have been made. The manuscript should be accepted.

Answer: Dear reviewer, we are incredibly grateful for your before remarks concerning the manuscript. We appreciate the time you spent editing it, emphasizing each point that will make the manuscript better. Thank you again for your contributions.

Round 2

Reviewer 2 Report (New Reviewer)

The revised version has adequately addressed most of the critiques raised by this reviewer and is now suitable for publication in "IJMS".

This manuscript is a resubmission of an earlier submission. The following is a list of the peer review reports and author responses from that submission.

Round 1

Reviewer 1 Report

Douglas Dourado et al., Resveratrol-loaded Attalea funifera oil organogel nanoparticles: a potential nanocarrier against A375 human melanoma cells. However, the following suggestions may improve the quality of the manuscript. I recommend its publication after making these necessary corrections.

-Advantages of organogel nanoparticles, marketed drug side effects, and nature drug treatment of melanoma (update reference).

-very important Characterization techniques, so author should be include FTIR, SEM, TEM and DLS analysis data.

-Add the particle size (Graph).

-All The statistics statistics information such as (Mean ± SD and statistical test used to calculate a p-value should be indicated in figure 1, 2 and 3. 

Minor editing of English language required

Reviewer 2 Report

Having read the manuscript I have the following comments:

1.  L2, L20 and elsewhere all Latin names should be written in italics such as  "Attalea funifera", please correct.

2. L95 what does SO and RSO refer to?

3. L114, what does 3.2.2 Zeta potential.... refer to?

4. L132 should 2.2.2 not be 2.2.3?

5.  Why was HaCaT and  FGH cells used as a non cancerous control for A375 melanoma cells?  Neither of these two cells are melanocytic in origin, so their comparative value is questionable.  FGH are dermal cells in origin and have a completely different physiology to epidermal cells. Please discuss in detail why these cells were chosen to be controls in this study.

6. L324-5 what was the final concentration of DMSO in the media containing the nanoparticles that were used in the experiments, was it 5% as stated?  

7. Please correct references 23, 25, 47.

The English needs some editing the use of phrases and that of italics require revision.